# Synthesis and Deposition of Ag Nanoparticles by Combining Laser Ablation and Electrophoretic Deposition Techniques

**Mònica Fernández-Arias [1], Massimo Zimbone [2],\*, Mohamed Boutinguiza [1], Jesús Del Val [1], Antonio Riveiro [1], Vittorio Privitera [2], Maria G. Grimaldi [2,3] and Juan Pou [1]**

1 Department of Applied Physics, University of Vigo, EEI, Lagoas-Marcosende, 36310 Vigo, Spain
2 Institute for Microelectronics and Microsystems, National Research Council (CNR-IMM), via S. Sofia 64, 95123 Catania, Italy
3 Dipartimento di Fisica e Astronomia, Università di Catania, via S. Sofia 64, 95123 Catania, Italy
\* Correspondence: massimo.zimbone@ct.infn.it; Tel.: +39-095-378-5350

**Abstract:** Silver nanostructured thin films have been fabricated on silicon substrate by combining simultaneously pulsed laser ablation in liquid (PLAL) and electrophoretic deposition (ED) techniques. The composition, topography, crystalline structure, surface topography, and optical properties of the obtained films have been studied by energy dispersive X-ray spectroscopy (EDS), high-resolution transmission electron microscopy (HRTEM), X-ray diffraction (XRD), and UV-visible spectrophotometry. The coatings were composed of Ag nanoparticles ranging from a few to hundred nm. The films exhibited homogenous morphology, uniform appearance, and a clear localized surface plasmon resonance (LSPR) around 400 nm.

**Keywords:** silver nanoparticles; electrophoretic deposition; pulsed laser ablation in liquid

## 1. Introduction

Noble metal nanoparticles in solution and deposited as thin films have attained wide popularity in the last 10 years and aroused intense research interest in nanotechnology due to their well-known properties, such as good conductivity, localized surface plasmon resonance (LSPR), and antibacterial and catalytic effects [1–4]. They are used in many different areas, such as medicine, solar cells, nano- and microelectronics, scientific investigations [5]. Films based on noble metals are the object of intense investigation due to the optical properties introduced by the characteristic LSPR, which generates an optical local field enhancement. This collective coherent oscillation of electrons on the conductive band of metallic nanoparticles interacts with the electromagnetic field and produces a strong absorption in particular regions of the electromagnetic spectrum. In particular, silver nanoparticles can absorb light in the near UV region of the spectrum extending their wavelength response to visible light. Other metal nanoparticles than noble metals also show absorption in visible light, such as Cu and Al [6,7], but their LSPR is weak and suffers from relatively easy oxidation. The most intensely studied plasmonic nanoparticle materials are Au and Ag. Ag is cheaper than Au and presents an even stronger LSPR, making Ag nanoparticles good candidates for many different applications. They can be used for enhancing solar cell efficiency by means of light trapping in organic as well as inorganic solar cells [8,9] or they can be used together with $TiO_2$ to improve its photocatalytic activity. These nanoparticles produce an increased absorption in the visible region due to their aforementioned LSPR [10]. The very high electric field around the nanoparticles makes them very good candidates for enhancing the signal of Raman scattering spectroscopy (SERS) [11,12], luminescence, and cathodoluminescence [13,14].

A wide range of techniques for producing nanoparticles is used in the literature, and can be roughly classified into two groups based on chemical or physical processes. The most popular method is chemical reduction, whereby organic or inorganic agents are used to reduce Ag salt and generate nanoparticles with different sizes, shapes, and composition [15–19]. Most of these methods use chemical reactants, surfactants, or stabilizers which contaminate the final nanoparticles, necessitating further extraction and purification [17]. It is worth pointing out that this is one of the main issues with nanoparticle technology. Physical processes, on the contrary, use metallic precursors and allow nanomaterials to be obtained without the presence of chemical reagents, but size and shape control is challenging. Among the physical techniques, laser-based ones have been used extensively. In particular, pulsed laser ablation in liquid (PLAL) is one of the most promising techniques to produce very pure nanoparticles. Indeed, it avoids surfactants and reaction products in solution. In PLAL, a high fluence laser pulse impinges on a solid substrate immersed in liquid. Plasma plume is formed and expands into liquid. The rapid cooling realizes nanoparticles in liquids. In previous works, we have obtained ceramic and metal nanoparticles in water and in open air [20–24]. Moreover, nanoparticles obtained by PLAL are charged and, once synthesized in solution, can be electrodeposited in a suitable substrate. There are different methods for nanoparticle deposition, such as chemical vapor deposition (CVD), physical vapor deposition (PVD), pulsed laser deposition (PLD), and sputtering. While most of these require sophisticated and expensive equipment [23–25], electrophoretic deposition (ED) is a simple and low-cost technique for nanomaterial deposition. This method has already been used to deposit Ag nanoparticles prepared by chemical methods [26]. It allows the synthesis parameters of nanoparticles to be controlled and incorporated into materials, which enables materials to be obtained and fabricated with tailored properties.

In this work, we report the results of combining the PLAL with ED technologies to obtain films of Ag nanoparticles deposited on a substrate. By combining a synthesis technique and a deposition method, Ag nanoparticles are produced and deposited in one step. This process will have great advantages in terms of costs and time, allowing a spread use of the combined technique.

## 2. Materials and Methods

Plates of Ag of 99.99% purity were cleaned and sonicated for ablation by laser in water. The targets were attached to the bottom of a glass box and filled with distilled water up to 1.0 mm over the upper surface of the Ag plate. The Ag targets were used as positive electrodes while a substrate of Si separated 2.5 cm was used as negative electrodes, as shown in Figure 1.

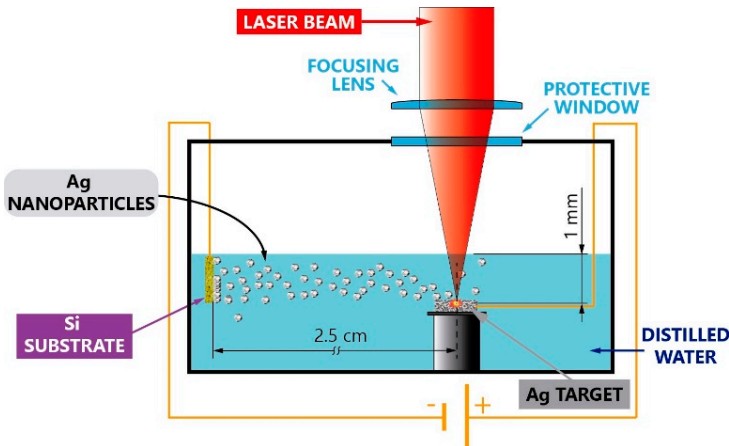

**Figure 1.** Schematic illustration of the experimental setup used to synthesize and deposit Ag nanoparticles on Si. Note: dimensions not to scale.

The laser source used consisted of a pulsed diode-pumped Nd:YVO4 laser (Rofin, Hamburg, Germany). It provides laser pulses at 532 nm with pulse duration of 14 ns, a repetition rate of 20 kHz,

and an average output power of 6.0 W. The Ag was ablated in water while an electric field of 15 V was applied between the electrodes.

Scanning electron microscopy (SEM) was used to observe and analyze the surface morphology and the microstructure by means of a JSM-6700 field emission scanning electron microscope (JEOL, Tokyo, Japan). Transmission electron microscopy (TEM), selected area electron diffraction (SAED), and high-resolution transmission electron microscopy (HRTEM) images were taken on a JEOL JEM-2010 FEG transmission electron microscope equipped with a slow digital camera scan, using an accelerating voltage of 200 kV, and provided by an energy dispersive X-ray spectrometer (EDS) to reveal their crystallinity and composition. The optical absorption of Ag nanoparticles deposited on the glass was measured by UV-Vis in a Hewlett Packard HP 8452 spectrophotometer (San Jose, CA, USA) in the wavelength range of 190–800 nm in a 10 mm quartz cell. Electron energy loss spectroscopy (EELS, JEOL, Tokyo, Japan) was carried out in order to check the potential presence of oxygen in the synthesized and deposited nanoparticles.

## 3. Results and Discussion

When a high-power nanosecond laser beam strikes on the Ag target, the local temperature rises above the Ag boiling point, leading to the formation of a plasma plume with the presence of different species such as atoms and ions. These species are confined by the surrounding liquid and remain inside the cavitation bubble realized by the evaporated solvent. Nanoparticles nucleate and grow by coalescence as the plasma cools down [27]. The obtained nanoparticles are charged and, due to the presence of an electric field inside the solution, can be accelerated and deposited on a suitable electrode, forming a thin film.

We analyzed both nanoparticles in solution and deposited on the substrate. Figure 2 exhibits the typical TEM image of the nanoparticles found in solution.

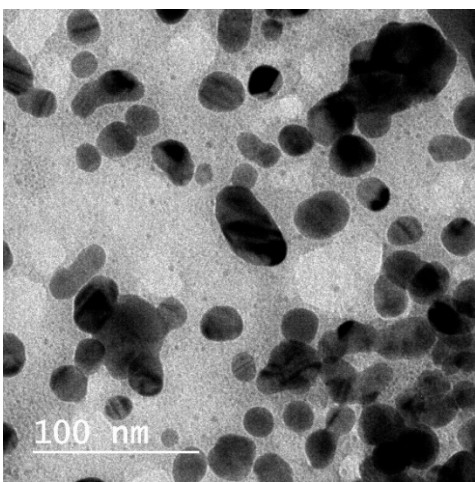

**Figure 2.** Transmission electron microscopy (TEM) micrograph of the obtained nanoparticles with an elongated shape.

The nanoparticles are ellipsoidal in shape. It is worth pointing out that they are not spherical as obtained in previous works [28,29]. The ellipsoidal shape of the nanoparticles is probably favored by the presence of the electric field, which contributes to aligning nanoparticles during their formation. Nanoparticles show broad distribution in terms of both size and shape. The agglomeration is probably produced during the deposition of the solution drop on the TEM grid.

Figure 3 shows the selected area electron diffraction (SAED) performed on different groups of particles. It shows the concentric diffraction rings with bright spots.

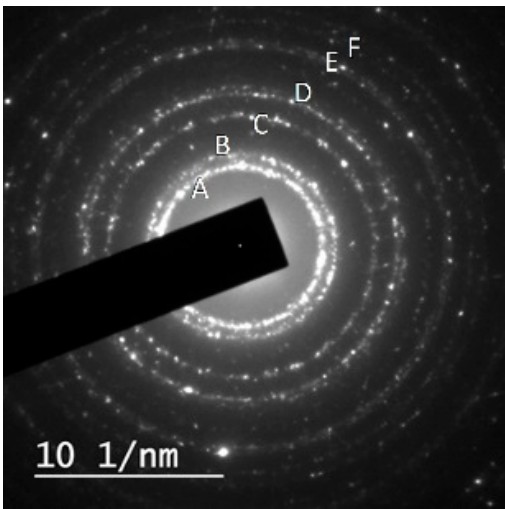

**Figure 3.** Selected area electron diffraction (SAED) pattern exhibiting diffraction rings obtained from Ag nanoparticles. The diffraction peaks from A, B, C, D, E, and F are consistent with Ag, showing 0.235, 0.204, 0.144, 0.123, 0.118, 0.101 nm and the corresponding {111}, {200}, {220}, {311}, {222}, {400} family planes of silver.

We observed that all the obtained particles are crystalline. The measured interplanar distances are listed in Table 1 and compared to those of metallic silver and $Ag_2O$.

**Table 1.** The d-spacing as measured from selected area electron diffraction (SAED) performed on Ag nanoparticles obtained by laser ablation of Ag target in water using 532 nm laser compared to those of metallic Ag and $Ag_2O$.

| $d_{hkl}$ (nm) | A | B | C | D | E | F |
|---|---|---|---|---|---|---|
| Measured | 0.235 | 0.204 | 0.144 | 0.123 | 0.118 | 0.101 |
| Ag JCPDS_ICDD (1993) | 0.236 | 0.204 | 0.145 | 0.123 | 0.118 | 0.102 |
| $Ag_2O$ JCPDS_ICDD (1993) | 0.237 | – | 0.143 | – | 0.118 | – |

The planar distances measured from the synthesized nanoparticles show good agreement with metallic silver even though the nucleation of the nanoparticle is performed in the presence of evaporated water. Their interplanar distances were calculated to be 0.235, 0.204, 0.144, 0.123, 0.118 and 0.101 nm, which could be assigned to the distances between {111}, {200}, {220}, {311}, {222} and {400} family planes of cubic silver.

To elucidate the crystalline nature of the nanoparticles more precisely, further investigations were made on single particles by means of HRTEM, as can be observed from Figure 4a, where clear lattice fringes with the d-spacing values are displayed.

The d-spacing measured from the corresponding fast Fourier transform (FFT) (Figure 4b) gives 0.205 and 0.238 nm which could be assigned to the {200} and {111} family planes of cubic silver whose corresponding interplanar distances are 0.204 and 0.236 nm respectively, while 0.238 nm could also be attributed to the {200} family planes of $Ag_2O$. The high magnification image in Figure 4a of the produced nanoparticle does not reveal any crystalline discontinuity or amorphous nature on the shell and no apparent oxide layer was found.

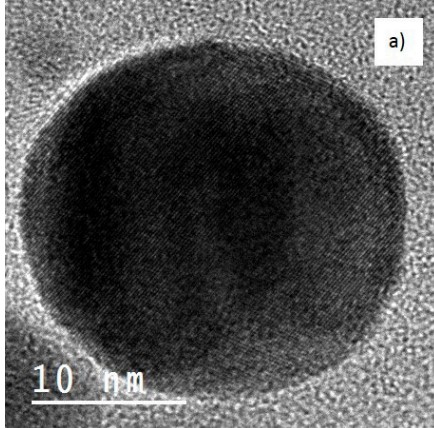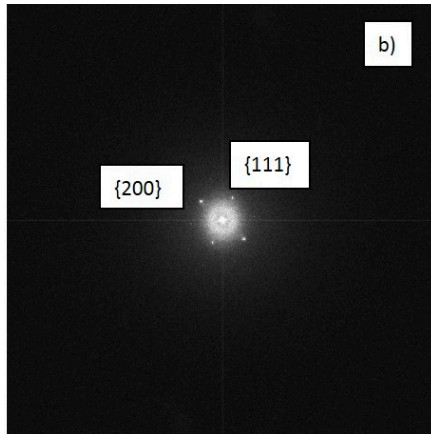

**Figure 4.** High-resolution transmission electron microscopy (HRTEM) of single particles showing clear fringes (**a**) with the corresponding fast Fourier transform (FFT) (**b**) whose d-spacing, 0.205 and 0.238 nm, could be assigned to the {200} and {111} family planes of cubic silver.

Together with the important specific area of the nanoparticles, their LSPR show dependence on their size and shape as well as the dielectric properties of the surrounding media [30,31]. The nanoparticles exhibit irregular shape rather than spherical morphology, which can favor the excitation of higher-order plasmon modes over small nanospheres. When the surface plasmon is excited, the intensity of the electromagnetic field around the sharp edges of nanoparticles can be significantly greater than the incident radiation [32]. Figure 5 shows the measured absorption spectrum of silver nanoparticles by laser irradiation of Ag submerged in water.

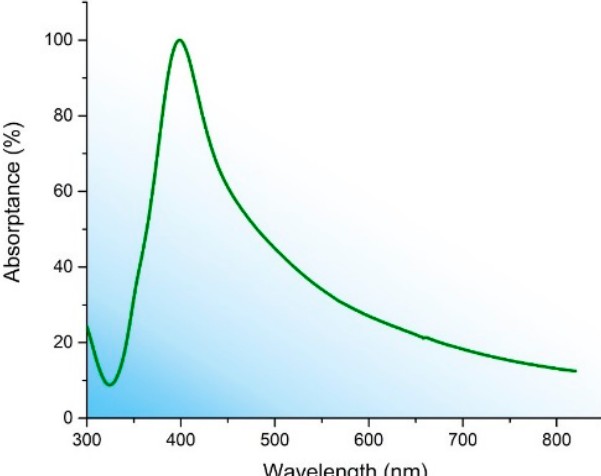

**Figure 5.** UV-Vis absorption spectrum of silver nanoparticles obtained by laser ablation of Ag target submerged in water with LSPR around 400 nm.

The UV-Vis absorption spectrum of the synthesized silver nanoparticles and grown-over Si substrate in the present work exhibits a plasmon resonance band around 400 nm, similar to characteristic LSPR of nanometric spherical particles [33]. The peak around 400 nm corresponds to dipolar mode excitation of spherical silver nanoparticles, but there is no significant position shifting of LSPR due to their size increasing from a few nanometers to tens of nanometers [34]. The large full width at half maximum (FWHM) is attributable to large polydispersity which promotes wider absorbance bands [35]. Although the nanoparticles synthesized by laser ablation in the solution can form stable colloidal solution, the nanoparticles can form networks or small assemblies [36] due to the attractive van der Waal forces in the absence of capping agents.

We now analyze the nanoparticles deposited on the Si substrate. For low electric fields (below 2.0 V/cm), no significant amount of particles collected on the cathode. When the voltage was increased to 15 V (approx. 6 V/cm), however, nanoparticle deposition on the cathode is observed. Figures 6 and 7 show SEM images of the deposited nanoparticles. It is worth noting that the final coating density, aspect, and shape are highly affected by the size and the ablated nanoparticles. To obtain a uniform coating on Si substrate it is important to limit the nanoparticles' size distribution, which is mainly related to the pulse duration and the laser power. Decreasing the pulse width or keeping the laser power (pulse energy) slightly above the ablation threshold leads to more uniform coatings. By increasing the deposition time, the coating density is increased but the uniformity is decreased.

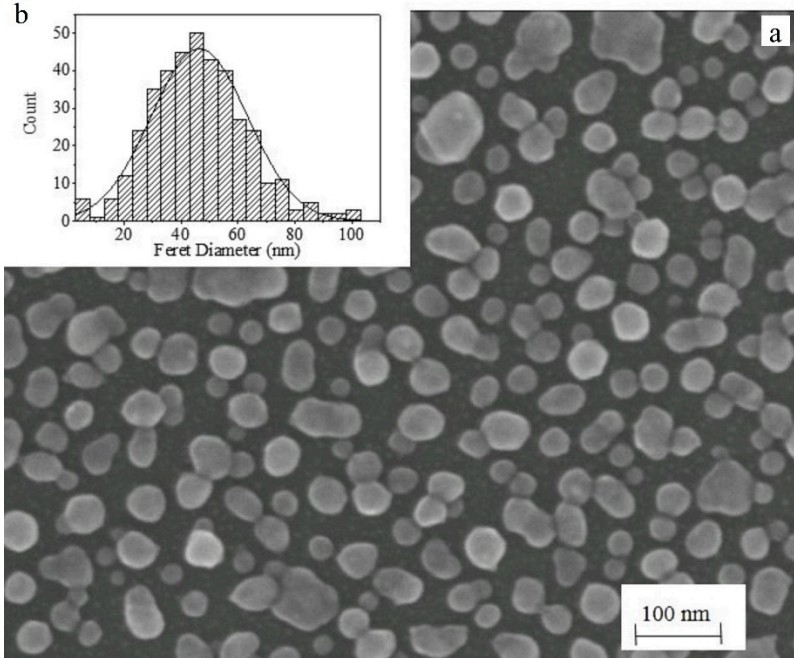

**Figure 6.** SEM micrograph (**a**) showing the distribution of the nanoparticles deposited on Si substrate with no agglomeration and the presence of slits among them. In (**b**), the Feret diameter distribution is shown.

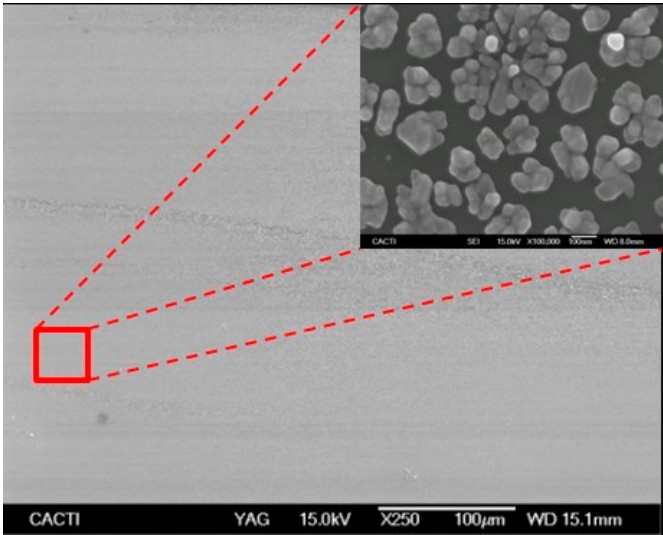

**Figure 7.** SEM micrograph showing the uniform film of the nanoparticles deposited on the Si substrate. The magnification in the inset illustrates the block formation on Si by the electric field.

We show the Feret diameter distribution in the inset of Figure 6. It is apparent that the size distribution on the cathode surface is uniform and monomodal. Monomodal distribution of nanoparticles can be explained by the Stokes law equation [37], which describes the mobility of small spherical nanoparticles in laminar flow under the action of electrical deriving force: $-v = qE/6\pi R\eta$. where $q$ is the particle charge, $E$ is the electrical field, $R$ is the nanoparticle hydrodynamic radius, $\eta$ is the water viscosity, and $v$ the nanoparticle velocity. Despite energy-dispersive size distribution and non-spherical shape of the nanoparticles, their $R/q$ ratio is very similar due to their reduced size, so the mobility and the distribution of nanoparticles on the substrate are mainly governed by the applied voltage, leading to a similar horizontal component velocity for particles. Gravity mainly affects bigger particles, deflecting their trajectories downward to not reach the cathode. On the other hand, the presence of an electrical field accelerates the charged nanoparticles to the cathode, contributing to the collision of particles of different sizes and the formation of assemblies. Most probably, the oversized blocks "sinking" before reaching the cathode.

The characteristics of the particles forming the obtained films are very important in terms of specific surface area and size, as well as their distribution on the substrate in order to enhance the efficiency of the incoming radiation, especially for peculiar applications such as photocatalysis or SERS. Indeed, the intensity of the electromagnetic field can be enhanced locally due to the interaction of plasmonic nanoparticles. Interaction enhances the electric field when the distance between nanoparticles is lower than the size of the particles. Indeed, strong electromagnetic enhancement can be detected in the gaps between nanoparticles when they are close to each other [38]. The deposited nanoparticles on Si, in this work, adopt the configuration mentioned above, showing slits among them smaller than the nanoparticle size, as can be seen from the SEM image in Figure 6. This fact makes the synthetized substrate suitable for SERS and photocatalytical applications.

## 4. Conclusions

In summary, we have synthesized silver nanoparticles and deposited them on Si substrate in a one-step process by combining the techniques of laser ablation of solids in liquids and ED to obtain uniform films. The silver target submerged in water was connected to the positive electrode and separated 2.0 cm from the cathode. The resulting nanostructured silver films consisted of homogenous coatings composed of irregular nanoparticles with slight slits among them. These films can be used to enhance the efficiency of photovoltaic devices, photocatalysis, or SERS.

**Author Contributions:** Conceptualization, Methodology, Validation and Writing—Original Draft Preparation: M.F.-A., M.Z., M.B., J.D.V. and A.R.; Writing—Review and Editing: V.P. and M.G.G.; Writing—Review, Editing and Supervision: J.P.

**Funding:** This work was partially supported by the EU research project Bluehuman (EAPA_151/2016 Interreg Atlantic Area), Government of Spain [RTI2018-095490-J-I00 (MCIU/AEI/FEDER, UE), Mobility Grant of Senior Professors and Researchers (Grant PRX15/00088)], and by Xunta de Galicia (ED431C 2019/23, ED481D 2017/010, ED481B 2016/047-0).

**Acknowledgments:** The technical staff from CACTI (University of Vigo) is gratefully acknowledged.

**Conflicts of Interest:** The authors declare no conflict of interest. The funders had no role in the design of the study; in the collection, analyses, or interpretation of data; in the writing of the manuscript, or in the decision to publish the results.

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
