# Peer review of "Synthesis and Deposition of Ag Nanoparticles by Combining Laser Ablation and Electrophoretic Deposition Techniques"

_coatings, doi:10.3390/coatings9090571_

Round 1

Reviewer 1 Report

The authors demonstrated a method on the synthesis and deposition of Ag particles. The results are interesting, but lack sufficient scientific understanding about the process. The following questions will help obtain a better understanding about the process and should be answered before this manuscript become publishable.

Authors should measure and show clearly the statistic size distribution of these Ag particles on substrates. A more complete study should be conducted on the Ag particle coating to obtain a better understanding about the process. Now the authors only demonstrated the deposition, but the understanding about the process is insufficient. For example, what experimental parameters are important for the process? For shorter or longer deposition time, will the Ag particle density on Si be changed? Will the uniformity be changed? Is the substrate (Si) important? What if the substrate is changed to glass/ITO, or metals? Following point 2, authors should study if the particle size can be changed by adjusting experimental parameters, such as the laser output power, repetition rate, etc.

Reviewer 2 Report

1. There are many papers about synthesizing AgNPs using wet chemistry method. I think it is better for authors to cite them in Introduction part, e.g., Ahn, Bok Y., et al. "Omnidirectional printing of flexible, stretchable, and spanning silver microelectrodes." Science323.5921 (2009): 1590-1593.  Shen, Wenfeng, et al. "Preparation of solid silver nanoparticles for inkjet printed flexible electronics with high conductivity." Nanoscale 6.3 (2014): 1622-1628.Raza, Muhammad, et al. "Size-and shape-dependent antibacterial studies of silver nanoparticles synthesized by wet chemical routes." Nanomaterials 6.4 (2016): 74.

2. According to Figure 2, size distribution of the AgNPs should be given.

3. In Figure 4b, {200} and {111} family planes should be marked. 

4. The AgNPs deposited on the Si substrate seems to be loose from Figure 7. Is it good for the following applications?

5. At least one application should be provided.

Reviewer 3 Report

This manuscript provides a one-step fabrication method combining laser ablation and electrophoretic deposition to produce nanostructured silver films on Si substrate. The manuscript, however, needs major revisions to be qualified to publish in Coatings. Below are the specific points that require attention.

The statement “… distilled water up to 2.0 mm over the upper surface of the Ag plate” is not the same as description in the figure 1. Please clarify for this. Why can the short pulse lasers make the ellipsoidal shape, instead of spherical shapes using CW and long pulse lasers. Please discuss about the size distribution of Ag nanoparticles What is the surface roughness and surface thickness of Ag film? How is the bonding strength of the coating film? Can the monomodal distribution of the produced ellipsoidal nanoparticles be explained by the Stokes law equation, which describes the mobility of small spherical nanoparticles, as presented in the manuscript? What are the results and the discussions of EDS and EELS measurements? There were several of grammatical mistakes. Please correct again.

Reviewer 4 Report

The paper deals with nanostructured thin films fabricated on silicon substrate by combining pulsed laser ablation and electrophoretic deposition. The concept is interesting and can be applicable in various industrial applications. A few more specific comments:

The audience of coatings might not be familiar with laser processing methods and where such methods can be used. I propose that the authors add a few sentences as an intro for the laser processing for example "lasers have been used for processing processing extensively etc." Suggested citations: A monolithic micro-tensile tester for investigating silicon dioxide polymorph micromechanics, fabricated and operated using a femtosecond laser, CE Athanasiou, Y Bellouard, Micromachines 6 (9), 1365-1386 and Seemingly unlimited lifetime data storage in nanostructured glass, J Zhang, M Gecevičius, M Beresna, PG Kazansky, Physical Review Letters 112 (3), 033901 Fig 5 the letters should be bigger. Overall, the article is interesting and I suggest it for publication.

Round 2

Reviewer 1 Report

The revised manuscript is publishable now.

Reviewer 3 Report

The manuscript is qualified to publish in Coatings.